# Determining the Most Suitable Ultrasound-Guided Injection Technique in Treating Lumbar Facet Joint Syndrome

**DOI:** 10.3390/biomedicines11123308

**Published:** 2023-12-14

**Authors:** Areerat Suputtitada, Jean-Lon Chen, Chih-Kuan Wu, Yu-Ning Peng, Tzu-Yun Yen, Carl P. C. Chen

**Affiliations:** 1Department of Rehabilitation Medicine, Faculty of Medicine, Chulalongkorn University, King Chulalongkorn Memorial Hospital, Bangkok 10330, Thailand; prof.areerat@gmail.com; 2Department of Physical Medicine & Rehabilitation, Chang Gung Memorial Hospital at Linkou, College of Medicine, Chang Gung University, Guishan District, Taoyuan City 33343, Taiwan; bigmac1479@gmail.com (J.-L.C.); xavierwu829@gmail.com (C.-K.W.); d422d422@gmail.com (Y.-N.P.); yenhsnu42@gmail.com (T.-Y.Y.)

**Keywords:** facet joint, lower back pain, dextrose, ultrasound, injection

## Abstract

(1) Background: Lower back pain is often caused by lumbar facet joint syndrome. This study investigated the effectiveness of three different injection methods under ultrasound guidance in treating elderly patients with lumbar facet joint syndrome. The difficulty in performing these injections was also evaluated; (2) Methods: A total of 60 elderly patients with facet joint syndrome as the cause of lower back pain were recruited and divided into 3 groups. Group 1 received medial branch block (MBB). Group 2 received intra-articular facet joint injections. Group 3 received injection into the multifidus muscle portion that covers the facet joint. Five percent dextrose water (D5W) was used as the injectant. The visual analog scale (VAS) was used to measure the degree of lower back pain; (3) Results: Before the injection treatments, the VAS score averaged about 7.5. After three consecutive injection treatments (two weeks interval), the VAS score decreased significantly to an average of about 1 in all 3 groups, representing mild to no pain. Between group analyses also did not reveal significant statistical differences, suggesting that these procedures are equally effective; (4) Conclusions: Ultrasound-guided injection of the multifidus muscle may be a feasible option in treating elderly patients with lower back pain caused by facet joint syndrome as it is easier to perform as compared to MBB and intra-articular facet joint injection.

## 1. Introduction

Lower back pain (LBP) is considered the most common pain syndrome in adults [1]. There are several causes of LBP, such as joint degeneration, muscle strain, and lumbar disc herniation [2]. The lumbar facet joint is the most common location causing pain and accounts for about 15–45% of LBP [3]. LBP caused by the facet joint is known as facet joint syndrome [3]. It is an arthritis-like condition, with the breakage of the cartilage between the inferior and superior articular processes, triggering pain signals to the innervated medial branch nerve endings. The most common site of facet joint syndrome is at the lumbar spine L4–L5 level [3].

However, the diagnosis of facet joint syndrome cannot be confirmed through patients’ history interviews or physical examinations. Springing and Kemp’s tests can elicit pain after extension from a fully flexed position and may provide evidence suggestive of facet joint syndrome [4,5]. A simple X-ray does not provide the information in confirming the diagnosis of facet joint syndrome, but may help in evaluating the degree of degeneration on the facet joints. When spurs are visible on the X-ray, it is highly likely that the degree of degeneration has reached an advanced level. Other imaging tools, such as computed tomography (CT) or magnetic resonance imaging (MRI) scans of the lumbar spine, may be used to diagnose this syndrome. However, none are specific to confirm the diagnosis of facet joint syndrome [4,6]. Diagnostic positive facet joint block or medial branch block (MBB) is a technique that can confirm that the facet joint is the cause of chronic spinal pain [7]. When symptoms of LBP disappear after the block, the diagnosis of facet joint syndrome can be confirmed. There are several injection techniques that can be applied in treating facet joint syndrome. However, due to spur formation and degenerative changes in elderly patients, needle insertion into the facet joints can be difficult [8]. In the lumbar spine, the medial branch has innervations to the facet joint and also to the multifidus muscle [9]. Accurate needle insertion to the medial branch to perform MBB relies on fluoroscopic or ultrasound guidance and can be technically difficult [10,11]. Musculoskeletal ultrasound can be the tool of choice in performing imaging guided injection procedures as it is radiation-free and offers real-time images [12]. On the other hand, multifidus muscle injection is easier to perform. Since the multifidus muscle is innervated by the median nerve and the muscle itself also covers the facet joint, intramuscular injection close to the facet joint may be effective in treating facet joint syndrome [13]. As a result, we hypothesize that injection into the multifidus muscle has a similar treatment effectiveness similar to that of MBB in the alleviation of pain caused by facet joint syndrome. The results obtained in this study provided us with evidence on the feasibility of a safer injection option that can be applied in treating LBP elderly patients caused by facet joint syndrome.

## 2. Materials and Methods

### 2.1. Participants

Sixty elderly patients with lower back pain (LBP) and with positive provocative results on the Kemp’s test and Springing test were recruited into this study. Facet joint syndrome was at the L4–L5, and L5–S1 levels. Patients with other causes of LBP, such as herniated intervertebral disc (HIVD), and traumatic back injury, were excluded from this study. These patients received 3 different injection treatments and were divided into 3 groups (20 patients in each group). This study was approved by the Institutional Review Board (IRB) of Chang Gung Medical Foundation and was conducted in accordance with the Declaration of Helsinki. All the patients signed the informed consent before participating in this study. The approved IRB number was: 202202069A3.

### 2.2. Treatment Protocols

Group 1 (n = 20): These patients received medial branch block (MBB) using musculoskeletal ultrasound guidance. With the patients placed in a prone position and a pillow under the abdomen, a low-frequency curvilinear transducer was placed in a transverse plane to obtain the transverse paravertebral sonogram of the lower back (Figure 1). Using the in-plane technique, the needle was inserted at an angle of about 45° to 60° to the skin, and in a lateral to medial direction to perform the medial branch block. The needle was directed to the bottom of the groove between the lateral surface of the superior articular process and the cephalad margin of the respective transverse process (Figure 1A,B). Once bony contact is felt, the transducer is then rotated 90° clockwise to obtain the longitudinal paravertebral sonogram to ensure that the needle tip is at the cranial edge of the transverse process (Figure 1C,D). Subsequent injection of 0.25 milliliters (mL) of D5W was performed. The injectant volume used in MBB cannot be of large volume as this may cause extravasation and false positivity of MBB [14]. 

Group 2 (n = 20): These patients received ultrasound-guided facet joint injection. With the patients placed in a prone position and a pillow under the abdomen, a low-frequency curvilinear transducer was placed in a transverse plane to obtain the transverse paravertebral sonogram of the lower back (Figure 2). Using the in-plane technique, the needle was inserted at an angle of about 45° to 60° to the skin, and in a lateral to medial direction to perform the facet joint injection. The needle was directed into the hypoechoic space resembling the facet joint. This facet joint is formed by the inferior articular, and superior articular processes. In some patients, high injection pressure was felt, and the injectant was difficult to inject into the facet joint. This may due to the fact that the facet joint is small and can only tolerate a small amount of injection fluid [15]. For facet joint injections, the recommended injectant volume is approximately 1.5 milliliters (mL). In this study, 2 mL injectant was prepared, and injection was terminated when resistance was encountered. 

Group 3 (n = 20): These patients received ultrasound-guided intramuscular multifidus muscle injection. With the patients placed in a prone position and a pillow under the abdomen, a low-frequency curvilinear transducer was placed in a transverse plane to obtain the transverse paravertebral sonogram of the lower back (Figure 3). Using the in-plane technique, the needle was also inserted at an angle of about 45° to the skin, and in a lateral to medial direction to perform the multifidus muscle injection. The needle was guided into the portion of the muscle that covers the facet joint. A total of 5 mL D5W was injected into this area as this is the recommended maximal volume of injectate that can be injected intra-muscularly [16] (Figure 3). 

The above procedures were performed in a rehabilitation outpatient clinic. Visual Analogue Scale (VAS) was used to measure the lower back pain intensity [17]. The numeric scale of 0 is considered as no pain, 1–3 as mild pain, 4–5 as moderate pain, 6–7 as severe pain, 8–9 as very severe pain, and 10 as excruciating pain [18]. Other data such as the duration of LBP, and body mass index (BMI) were also recorded. At an interval of 2 weeks apart, a total of 3 injections were performed for each patient. VAS scales were measured before, during, and 1 and 3 months after the completion of injection treatments. VAS scale was assessed for each patient at the time of every outpatient clinic visit. The same physician acquainted with ultrasound images performed all the MBB and injection procedures to avoid experimental bias. From the obtained sonoanatomy images, the physician can detect the correct level of facet joints and transverse processes. The T3300 Tablet Ultrasound Imaging System with a bandwidth of 2–6 MHz C62B Convex Array curvilinear transducer was used (BenQ Medical Technology Corporation, Taipei, Taiwan) to perform all the MBB and injection procedures. 

### 2.3. Statistical Analysis

Data obtained in this study were tested for normality using the Kolmogorov-Smirnov test. Results were expressed as mean ± standard deviation (SD) in the tables. In the figure, lines were drawn as mean ± standard error of the mean (SEM). The between group VAS score comparison before any injection treatments was analyzed using Kruskal–Wallis non-parametric analysis of variance (ANOVA) followed by Dunn’s post-hoc test. For within group comparisons of VAS scores between the different time periods, the analytical tool of Friedman’s repeated measures ANOVA on ranks followed by Dunn’s post-hoc test was applied. Repeated measures ANOVA was used to assess other variables such as BMI, age, and disease duration. Descriptive statistics were presented with means and standard deviations (SD) for normally distributed continuous variables and median and interquartile range (IQR) for non-normally distributed continuous variables. The SPSS analytical software was used (version 21.0, SPSS Inc., Chicago, IL, USA). Statistical significance was set at a *p* value of less than 0.05.

The G-Power 3 software, for determining the minimal sample size, was used in the group randomization selection process. The significance level (α) was set at 0.05 and a power (1 − β) of 0.80, and a total of 3 groups along with 6 times repeated measurements. Then the minimal sample size was calculated to be 60, and the estimated minimum patient number in each group was 20. The method of randomly permuted blocks was accessed on 6 June 2020 for patient recruitment and in deciding which group the patient will be enrolled: http://www.jerrydallal.com/random/assign.htm.

## 3. Results

By applying the Kolmogorov–Smirnov test, the VAS data were not normal distributed. On the other hand, the demographic and clinical data of age, BMI, and disease duration were normally distributed. These data did not reveal significant statistical differences in between group comparisons (Table 1).

In group 1, the median (IQR) of VAS score before MBB was measured to be 8 (IQR 2), indicating severe pain. After the first MBB, VAS score was significantly decreased to 3 (IQR 1) when measured at 2 weeks later. At this time, second MBB was performed, and the VAS score was measured to be 2 (IQR 0) 2 weeks later, and IQR = 0 means almost VAS = 2. Then, the third and final MBB was performed, and VAS score was 0.5 (IQR 1) when measured at 2 weeks later, indicating mild to nearly no pain. At 3 months after the completion of all MBB procedures, VAS score was measured to be 4 (IQR 2), indicating mild to moderate pain, which was still significantly less as compared to the VAS score before MBB (*p* < 0.05) (Table 2 and Figure 4). 

In group 2, the mean VAS score before the facet joint injection was measured to be 7 (IQR 1.8), indicating severe pain. After the first facet joint injection, the VAS score was significantly decreased to 5 (IQR 0), and IQR = 0 means almost VAS = 5 when measured at 2 weeks later. At this time, the second facet joint injection was performed. Two weeks later, the VAS score was measured to be 2 (IQR 1). Then the third and final facet joint injection was performed, and the VAS score was 1 (IQR 1) when measured at 2 weeks later, indicating mild to nearly no pain. At 3 months after the completion of all facet joint injections, the VAS score was measured to be 4 (IQR 1), indicating mild to moderate pain, which was still significantly less as compared to the VAS score before facet joint injection (*p* < 0.05) (Table 2 and Figure 4). 

In group 3, the mean VAS score before the multifidus muscle injection was measured to be 7 (IQR 1.8), also indicating severe pain. After the first multifidus muscle injection, the VAS score was significantly decreased to 4 (IQR 2) when measured at 2 weeks later. At this time, the second multifidus muscle injection was performed. At a follow up period 2 weeks later, the VAS score was measured to be 2 (IQR 0.8). Then the third and final multifidus muscle injection was performed, and the VAS score was 1 (IQR 1) when measured at 2 weeks later, indicating mild to nearly no pain. At 3 months after the completion of all multifidus muscle injections, the VAS score was measured to be 5 (IQR 2), indicating mild to moderate pain, which was still significantly less as compared to the VAS score before multifidus muscle injection (*p* < 0.05) (Table 2 and Figure 4).

## 4. Discussion

It is crucial to find the correct cause of LBP in order to apply the most suitable treatment option. Lumbar facet joint syndrome is the most frequent cause of LBP. However, there are no specific imaging tool or physical examination tests that can 100% confirm the diagnosis of facet joint pain syndrome. The lumbar facet joint syndrome referral pain symptoms are mainly located at the lower back, buttocks, and lateral thigh areas. Positive facet joint block or MBB is perhaps the most reliable method in confirming the diagnosis of lumbar facet joint pain syndrome. Injecting hyaluronic acid (HA), autologous platelet rich plasma (PRP), amniotic membrane/umbilical cord particulate (ex/micronized dehydrated human amnion/chorion membrane (μ-dHACM)) or steroids into the facet joint may offer treatment benefits [19,20,21,22,23]. However, which injectant offers the best effectiveness remains controversial [24]. This may be due to the fact that the facet joint is small and can only tolerate a small amount of injection fluid [15]. For facet joint injections, the recommended injectant volume is approximately 1.5 milliliters (mL). In this study, 2 mL injectant was prepared, and injection was terminated when resistance was encountered. Forceful injection would result in joint capsule rupture and further exacerbation of pain [25].

The facet joint can be difficult to inject in the elderly population as spurs and cartilaginous metaplasia may impede successful needle insertion even under ultrasound or fluoroscopic guidance (Figure 2A) [26]. Injecting large volumes of injectants may cause capsular damage [27]. Continuing the injection procedure may cause the dispersion of the injectant to the nearby facet joint soft tissue structures, including the medial branches and the epidural space. As a result, studies have suggested that facet joint injections may not be the suitable option in treating facet joint syndrome. Instead, precise MBB is the treatment of choice. Not only can MBB confirm the diagnosis of facet joint syndrome, subsequent neurolysis using radiofrequency or cryoablation can then be performed to the medial branches to ensure better treatment outcome [3,28].

MBB cannot be performed blindly or under palpation-guided needle insertion. Accurate MBB relies on ultrasound or fluoroscopic guidance. Musculoskeletal ultrasound is the tool of choice as compared with fluoroscopy in performing imaging guided injection procedures as it is radiation-free and offers real-time images. As a result, ultrasound-guided MBB can be performed in an outpatient setting while fluoroscopic guided injection needs to be done in an operating room (Figure 1) [29]. For maximum accuracy, all the nerve block and injection procedures in this study were performed under ultrasound guidance [12]. Physicians acquainted with sonoanatomy can easily locate the correct level of medial branch. However, ultrasound-guided MBB is technically dependent. The ascending and descending medial branches of the spinal nerve dorsal ramus innervate the facet joints. The medial branch courses under the collagenous slip of the mamillo-accessory ligament (MAL). The ascending branch is located cranially to the MAL [30]. As a result, if the injection needle is not accurately placed cranially to the MAL, ineffective blocking of the ascending branch can occur. In this study, MBB was accurately performed under the prone position and with a pillow placed under the abdomen (Figure 1). The injectant volume used in peripheral nerve block cannot be of large volume as this may cause extravasation and false positivity of MBB [14]. As a result, a volume of 0.25 mL 5% dextrose water (D5W) was used in this study to perform MBB. 

Multifidus muscle originates from the sacrum, erector spinae aponeurosis, posterior superior iliac spine, and inserts onto the spinous process. It is the most medial paraspinal muscle and plays an important role in intervertebral stability. The multifidus muscle is innervated by the medial branch nerve of the posterior ramus of the spinal nerve, which exits the spinal canal superior-lateral to the facet joint [9]. The facet joint is covered by the multifidus muscle. As a result, when the needle is inserted into the muscle portion that covers the facet joint, the injectant may adequately bathe the facet joint and the innervated medial branches. In this study, 5 mL of D5W was used because this is the recommended maximal volume of injectate that can be injected intra-muscularly [16]. Results in this study have shown that the injection of 5 mL D5W into the multifidus muscle has a similar treatment effectiveness similar to that of MBB in the alleviation of pain caused by facet joint syndrome. The rationale behind this is that the medial branches innervating the facet joint are bathed by the injected D5W, offering similar treatment effect as compared to that of MBB. The application of ultrasound-guided multifidus muscle is less technically demanding. Under the transverse view, the needle can be easily guided to the portion of the multifidus muscle that covers the facet joint (Figure 3A,B). Injection of 5 mL D5W can be contained within the multifidus muscle and will not cause the dispersion of the injectate into other soft tissue structures. 

The strategy of perineural injection using D5W has been shown to be effective in treating patients with carpal tunnel syndrome (CTS) and obturator neuralgia [31,32]. Injecting steroids to the nerve is not recommend due to possible neurotoxicity. As a result, D5W can be a suitable injectant in performing nerve blocks [33]. Intra-articular dextrose water injection has shown better outcome as compared to steroid when treating sacroiliac joint pain [34]. In this study, D5W was shown to be an appropriate injectant. The proposed mechanism is the dextrose mediated inhibition of the transient receptor potential cation channel subfamily V member 1 (TRPV1) receptors, and neurogenic inflammation [35]. Studies have shown that nerve block using D5W to bath the genicular nerves can offer effective pain reduction lasting from 4 hours to several weeks in patients with knee osteoarthritis [32,36]. D5W has an osmolality similar to the human physiological condition, and is not harmful to the nerves [32]. The injection of perineural D5W has shown to offer safe and outstanding long-term effects in the treatment of CTS [33]. 

There were shortcomings in this study. For example, different injectants were not used. The injection of PRP or steroids may be injected to the medial branch, facet joint, and multifidus muscle to observe the treatment effectiveness as compared with D5W. However, intramuscular injection of HA is not recommended, and should not be used as a feasible injectant option. In future studies, a larger volume of D5W may be injected into the multifidus muscle, such as 30 mL [37]. By placing the transducer in an in-plane relationship with the multifidus muscle, D5W can be injected into the muscle that covers several levels of the lumbar vertebrae [38]. This may be equivalent to performing MBB to several levels of the lumbar spine. 

## 5. Conclusions

Physicians acquainted with sonographic interpretation and ultrasound-guided injection skills are required to perform this procedure. Ultrasound-guided needle insertion into the multifidus muscle close to the facet joint is easier to perform as compared with MBB and facet joint injections. Inserting the needle to the cranial edge of the transverse process in performing MBB can be technically demanding. Facet joints with spur formation may impede successful needle insertion. Results obtained in this study suggested that ultrasound-guided multifidus muscle injection using 5% dextrose water can be a feasible treatment choice in alleviating lower back pain symptoms caused by facet joint syndrome. 

## Figures and Tables

**Figure 1 biomedicines-11-03308-f001:**
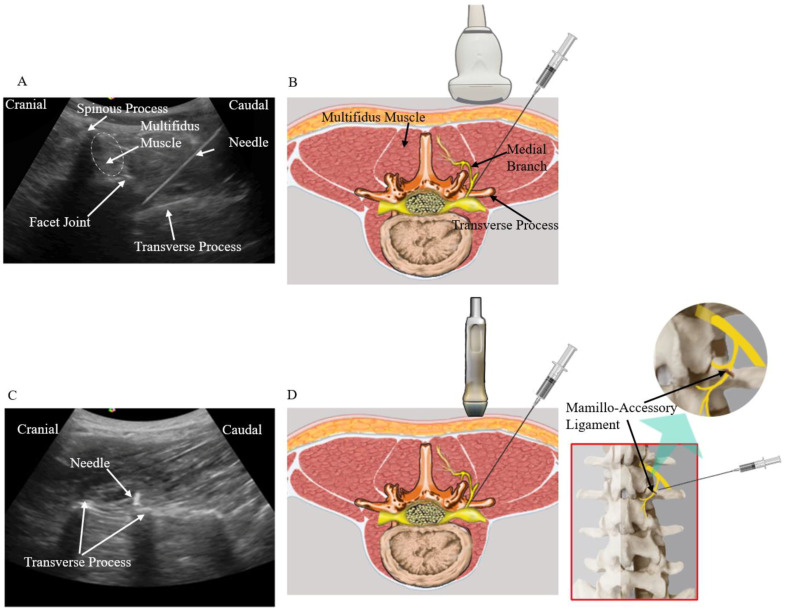
(**A**) Sonographic cross axis view of the lower back and its marked soft tissue structures. (**B**) A low-frequency curvilinear transducer is used. The multifidus muscle, facet joint, and the transverse process can be simultaneously observed. The needle is inserted at an angle of about 45 degrees in a lateral to medial direction and directed to the junction between the facet joint (superior articular process) and the superior border of the transverse process. (**C**,**D**) The transducer is then rotated 90 degrees to obtain the long axis view of the transverse process to confirm that the needle tip is located at the cranial edge of the transverse process. (**D**) The location of the mamillo-accessory ligament.

**Figure 2 biomedicines-11-03308-f002:**
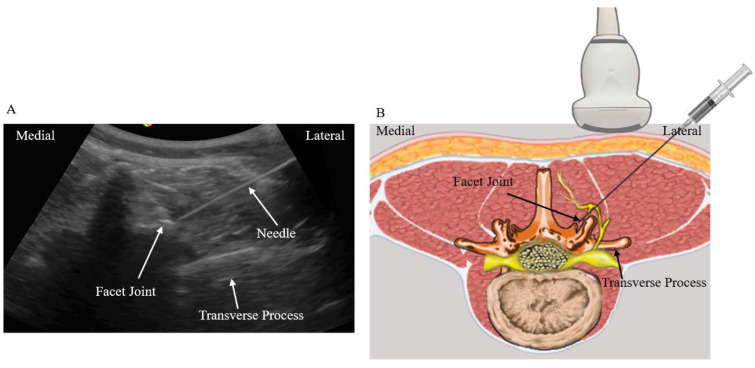
(**A**) Sonographic cross axis view of the lower back and the location of the facet joint. As observed in the figure, the superior articular process has a large bony spur. Needle insertion into the facet joint is not possible. (**B**) The needle is inserted at an angle of about 45 degrees in a lateral to medial direction and directed to the facet joint.

**Figure 3 biomedicines-11-03308-f003:**
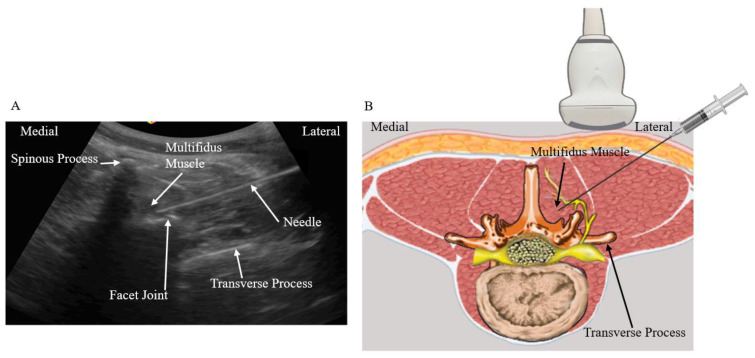
(**A**) Sonographic cross axis view of the lower back and the location of the multifidus muscle. The needle is directed into the area of the muscle that covers the facet joint. (**B**) The needle is inserted at an angle of about 45 degrees in a lateral to medial direction and directed into the multifidus muscle.

**Figure 4 biomedicines-11-03308-f004:**
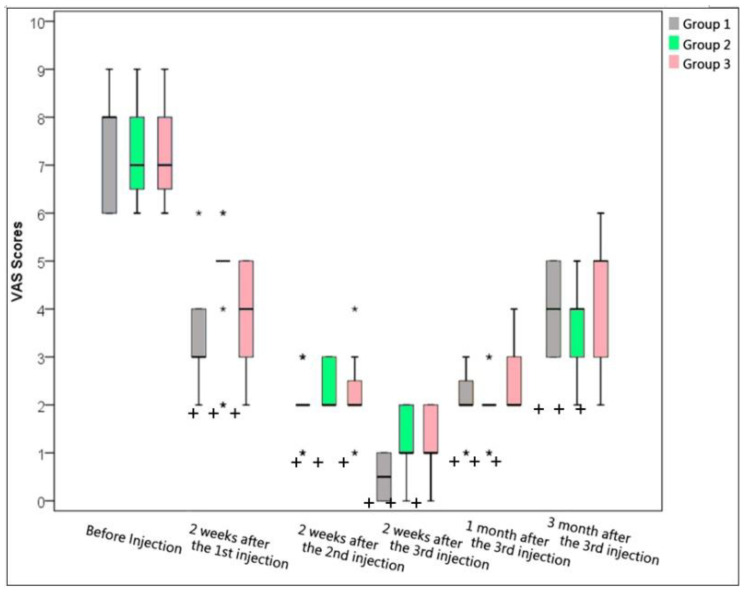
The comparison of VAS scores (0–10) among three groups. VAS box plots were drawn for each group (Group 1: Medial Branch Block; Group 2: Facet Joint Injection; Group 3: Multifidus Muscle Injection). The horizontal lines represent the medians and quartiles. The top and bottom of the vertical lines specify 1.5 times the interquartile range plus and minus the upper and lower quartiles, respectively. Values above and below the vertical lines were plotted with asterisks. “+” indicated statistically significance with *p* < 0.05 as compared with before injection in all three groups.

**Table 1 biomedicines-11-03308-t001:** Demographic and clinical characteristics of the recruited patients.

Characteristics	Group 1 (n = 20)	Group 2 (n = 20)	Group 3 (n = 20)	*p*
Age, years	71.3 ± 6.4	72.6 ± 5.7	73.7 ± 6.5	0.38
Sex (male/female)	12/8	11/9	11/9	
BMI (kg/m^2^)	25.0 ± 1.9	24.5 ± 1.8	25.8 ± 2.2	0.32
VAS scores (initial) ^a^	8 (IQR 2)	7 (IQR 1.8)	7 (IQR 1.8)	0.54 ^b^
Disease duration (Weeks)	51.8 ± 24.2	61.3 ± 29.2	49.9 ± 20.2	0.20

Values are expressed as mean ± standard deviation (SD) or sample size in numbers (*n*). VAS scores ^a^ are expressed as median (IQR). Wilcoxon Rank Sum test was used to test non-normally distributed continuous variables for statistical differences ^b^. Abbreviations: BMI, body mass index; VAS, visual analogue scale; Group 1: Medial Branch Block; Group 2: Facet Joint Injection; Group 3: Multifidus Muscle Injection. kg: kilograms. m: meters.

**Table 2 biomedicines-11-03308-t002:** The comparison of VAS scores between groups.

VAS (0–10), Median (IQR)	Group 1	Group 2	Group 3	*p* Value between Group Comparisons
Before injection (time of the first injection)	8 (IQR 2)	7 (IQR 1.8)	7 (IQR 1.8)	*p* = 0.54
2 weeks after the first injection (time of second injection)	3 (IQR 1)	5 (IQR 0)	4 (IQR 2)	*p* = 0.50
* *p* value (vs. before injection)	<0.05	<0.05	<0.05	
2 weeks after the second injection (time of third injection)	2 (IQR 0)	2 (IQR 1)	2 (IQR 0.8)	*p* = 0.94
* *p* value (vs. before injection)	<0.05	<0.05	<0.05	
2 weeks after the third injection	0.5 (IQR 1)	1 (IQR 1)	1 (IQR 1)	*p* = 0.95
* *p* value (vs. before injection)	<0.05	<0.05	<0.05	
1 month after the third injection	2 (IQR 0.8)	2 (IQR 0)	2 (IQR 1)	*p* = 0.81
* *p* value (vs. before injection)	<0.05	<0.05	<0.05	
3 months after the third injection	4 (IQR 2)	4 (IQR 1)	5 (IQR 2)	*p* = 0.83
* *p* value (vs. before injection)	<0.05	<0.05	<0.05	

Values are expressed as median and interquartile range (IQR). Group 1: Medial Branch Block. Group 2: Facet Joint Injection. Group 3: Multifidus Muscle Injection. VAS: Visual Analog Scale. Vs.: Versus. * = Statistically significant as the *p* value is <0.05.

## Data Availability

Table 1 and Table 2 have detailed data and statistical analysis of this study. We are willing to disclose any further data inquired by the readers.

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
