# Peer review of "Determining the Most Suitable Ultrasound-Guided Injection Technique in Treating Lumbar Facet Joint Syndrome"

_biomedicines, 2023, doi:10.3390/biomedicines11123308_

Round 1
Reviewer 1 Report
Comments and Suggestions for Authors
Dear Authors
Thank you for your work
I have some questions regarding your paper, specifically:
-Line 56-57: A reference is needed
-How the patients were allocated in one group? Did you randomize allocation?
-Why did you choose to inject 2ml? Facet jpint volume is approximately 0.8 ml, using higher volumes cound strecth the jint capsule and exacerbate pain
-VAS scores were not normally distributed but you used mean values, median and interquartile range should be calculated instead
-I would add a graphic representing table 2 data, it would help the readers
Author Response
-Line 56-57: A reference is needed
- Thank you for the suggestion. Reviewer 1 was correct, adding a reference to lines 56 -57 will be helpful. Therefore, a reference is now added documenting that multifidus muscle covers the facet joint.
- The Lumbar Multifidus Muscles are Affected by Medial Branch Interventions for Facet Joint Syndrome: Potential Problems and Proposal of a Pericapsular Infiltration Technique. J Gossner. AJNR Am J Neuroradiol. 2011 Dec;32(11):E213. doi: 10.3174/ajnr.A2901.Epub 2011 Nov 3.
-How the patients were allocated in one group? Did you randomize allocation?
- Sincere apologies for not mentioning group allocation clearly. In this study, the G-Power 3 software was applied in group randomization selection. This is now clearly mentioned in the statistical analysis section.
-Why did you choose to inject 2ml? Facet jpint volume is approximately 0.8 ml, using higher volumes cound strecth the jint capsule and exacerbate pain
- Yes, we are highly aware of this. We apologize that this was not clearly described in the text. In this study, 2 mL of injectant was prepared. Injection was ceased when resistance or high injection pressure was experienced as redundant injection may result in capsule destruction or other consequences. This is now clearly mentioned in the Method and Discussion sections.
-VAS scores were not normally distributed but you used mean values, median and interquartile range should be calculated instead
- Sincere apologies for not presenting the data well. We have recalculated the data, and have now presented the data using median and interquartile range. These are now clearly mentioned in tables 1 and 2.
-I would add a graphic representing table 2 data, it would help the readers
- Thank you for the comment. A figure is now drawn for table 2. We have spent more than 1 week drawing this figure. I hope reviewer 1 can be satisfied with the figure.
Reviewer 2 Report
Comments and Suggestions for Authors
There is an interesting paper, with important topic: the lumbar spine injections. However, there are some questions/ comments:
1. ultrasound-guided injection or under fluoroscopy - I need more details, pros and cons. the choice of ultrasound should be clearly stated, from introduction to conclusions.
2. the volume of liquid - why authors chose such (different in different places): it should be commented in methods and in discussion.
3. 5%dextrose - authors should describe why they chose it (a little bit longer, with references).
Author Response
- ultrasound-guided injection or under fluoroscopy - I need more details, pros and cons. the choice of ultrasound should be clearly stated, from introduction to conclusions.
- We want to thank reviewer 2 for this precious comment. It is now mentioned that ultrasound is radiation-free and offers real-time images as compared with fluoroscopy. It is now clearly stated in the introduction and in the discussion sections that ultrasound is the preferred imaging of choice.
- the volume of liquid - why authors chose such (different in different places): it should be commented in methods and in discussion.
- Thank you for the comment. It is now mentioned clearly in the Method and Discussion sections as to the amount of injectants that are applied in each injection procedure.
- 5%dextrose - authors should describe why they chose it (a little bit longer, with references).
- 5% dextrose is now widely used in rehabilitation medicine, especially in the treatment of carpal tunnel syndrome. Several papers are published regarding the treatment of carpal tunnel syndrome using 5% dextrose water. Another reference is now added in the Discussion section.